# Interrogating the Diversity of Vaginal, Endometrial, and Fecal Microbiomes in Healthy and Metritis Dairy Cattle

**DOI:** 10.3390/ani13071221

**Published:** 2023-03-31

**Authors:** Taurai Tasara, Anja Barbara Meier, Joseph Wambui, Ronan Whiston, Marc Stevens, Aspinas Chapwanya, Ulrich Bleul

**Affiliations:** 1Institute for Food Safety and Hygiene, Vetsuisse Faculty, University of Zurich, 8057 Zurich, Switzerland; taurai.tasara@uzh.ch (T.T.); mstevens@fsafety.uzh.ch (M.S.); 2MSD Animal Health Switzerland, 6005 Lucerne, Switzerland; anja.barbara.meier@msd.com; 3Department of Clinical Sciences, Ross University School of Veterinary Medicine, Basseterre 00265, Saint Kitts and Nevis; rwhiston@rossvet.edu.kn (R.W.); achapwanya@rossvet.edu.kn (A.C.); 4Department of Farm Animals, Clinic of Reproductive Medicine, Vetsuisse Faculty, University of Zurich, 8057 Zurich, Switzerland

**Keywords:** bacteria, metritis, RNA sequencing, microbiome, bovine, endometrium, vagina, uterus

## Abstract

**Simple Summary:**

Research on the microorganisms in the reproductive tract of cows has become increasingly popular. Reproductive pathogens, including bacteria, caused uterine disease and decrease fertility. Using sequencing techniques endometrial microbiomes in healthy animals and those with metritis were compared. Our study has identified uterine microbiome profiles that are positively and negatively associated with uterine health. Since it is important to know which bacteria live in healthy or diseased animals, this information will enable the development of treatment options for cows that not only reduce antibiotic use but improve fertility. An improved understanding of changes to the bacteria communities will help to identify animals that can successfully become pregnant again after calving.

**Abstract:**

The bovine genital tract harbors a dynamic microbiome. Genital tract microbial communities in healthy animals have been characterized using next-generation sequencing methods showing that microbe compositions differ between the vagina and uterus, more so during the postpartum period. Pre-calving fecal and vaginal, and endometrial swabs at the different postpartum intervals were collected from dairy cows. Microbiomes in these samples were determined based on bacterial 16S amplicon sequencing and compared between healthy (H; *n* = 10) control animals and cows that developed metritis (M; *n* = 10) within 21 days postpartum (DPP). Compared to healthy animals the pre-calving fecal and vaginal microbiomes of metritis animals were more abundant in sequences from the phylum Fusobacteria and the bacterial genera such as *Escherichia-Shigella* and *Histophilus*. In addition, compared to healthy animals, metritis cows harboured low microbial species diversity in the endometrium, as well as decreasing Proteobacteria and increasing Fusobacteria, Firmicutes, Actinobacteria, and Bacteroidetes abundances. The greatest taxonomic compositional deviations in endometrial microbial communities between the metritis and health cows were detected between 7 and 10 DPP. There was high taxonomic similarity detected between postpartum endometrial microbiomes and the prepartum vaginal and fecal microbiomes suggesting that colonization through bacteria ascending from the rectum and vagina to the uterine cavity might play a major role in establishing the endometrial microbiome postpartum. A deeper understanding of the establishment and dynamics of postpartum endometrial microbial communities in cows will thus provide crucial basic knowledge to guide the development of genital microbiome manipulation strategies for preventing uterine disease and improving fertility in dairy cows.

## 1. Introduction

High reproductive efficiency in the dairy cow requires a disease-free postpartum period. Endometrial and vaginal microbiomes are critical in the study of endometritis, which is an important cause of infertility in cattle. Following calving, a common characteristic of uterine involution is microbial influx into the endometrium. Thus, in addition to the increasing energy demands for lactation, a dampened immune status may lead to uterine infection in approximately 40% of the animals postpartum [1]. The resultant uterine diseases perturb fertility [2,3,4]. Cows with metritis suffer from infection and inflammation of the uterus while endometritis is an inflammation of the superficial layer of the uterus. Clinically, a distinction is made between puerperal metritis, which usually occurs in the first 10 days after birth and is associated with general clinical symptoms and fetid discharge, and clinical endometritis, which is characterized by (muco)purulent vulval discharges [1,5]. There exists a relationship between scoring uterine secretions and the abundance of microbes present in the uterus identified using culture-dependent methods [6,7,8]. However, given that bovids possess a complex yet dynamic microbiome, only a small proportion of the bacteria present in the uterus are detected using this approach [9]. Endometrial and vaginal microbiomes are critical in the study of endometritis, which is an important cause of infertility in dairy cattle. Thus, culture-independent approaches to unravel endometrial microbiomes are warranted [2]. Previously, metagenomic sequencing of the 16S rRNA genes in cattle with or without metritis, described 28 different phyla, including Bacteroidetes, Proteobacteria, Fusobacteria, and Firmicutes [10]. Compared to healthy animals, cows with metritis harbored more abundant Bacteroidetes. At the genus level, there were 824 genera detected, with *Fusobacterium*, *Bacteroides*, *Coxiella*, and *Porphyromonas* being the most abundant [10]. The genera *Fusobacterium* and *Bacteroides* are highly abundant in animals with copious vulval discharges and severe uterine disease. Similar bacterial communities were found in the vagina and postpartum uterine secretions; however, it was not possible to predict the occurrence of metritis from the vaginal microbiome [11]. In other studies, quantitative qPCR did not show a link between a diagnosis of metritis and the detection of *Escherichia coli* and *Truepurella pyogenes* [12]. This is contrary to other studies where these pathogens were frequently isolated from cattle with metritis [6,13]. The origins of the various pathogens are not yet fully known. Although the uterus is not free of bacteria even during pregnancy [14,15], it is thought that endometrial microbiota originates from the gut, the vagina, and the environment [1]. This study aimed to interrogate vaginal, endometrial, and fecal microbiomes in dairy animals and determine whether these profiles predict the risk of uterine disease postpartum.

## 2. Materials and Methods

### 2.1. Examined Animals

On a dairy farm with 100 Holstein Frisian cows, a total of 39 animals that calved were examined over a 9-month period. All cows calved spontaneously or with minor traction support from a maximum of one person. The animals were clinically examined on days 1, 4, 7, 10, and 21 DPP. There were no placenta retentions recorded. All animals were housed on hay bedding and fed Total Mixed Ration (TMR) comprising maize, grass silage, hay, and concentrate. The animals were clinically examined and their general condition, rectal temperature, heart and respiratory rate, mucous membranes, and gastrointestinal tract were assessed prior to sampling. Then, palpation per rectum and vaginal exploration were performed. Uterine size, the opening of the cervix, and cervical secretion were assessed and scored as previously described [16]. Briefly, vulval discharges were scored on a scale from 1 to 5, where: 1 is clear or translucent mucus; 2 is mucus containing flecks of yellowish white coloured pus; 3 is discharge containing ≤50% mucopurulent material; 4 is discharge containing >50% purulent material; and 5 is foul-smelling watery, reddish, or brownish discharge [17]. Animals showing signs of poor general health status and fever (>39.5 °C) in addition to an enlarged uterus and watery, foul-smelling discharge within 10 days postpartum (DPP) were deemed puerperal metritis animals [18,19]. Animals with a distended uterus, and purulent vulval secretions, but no signs of clinical illness 21 DPP were classified as clinical metritis cows. After classification, animals that received antibiotic treatment were excluded from the study (*n* = 19). The 20 animals selected for microbial analysis, were grouped into metritis (*n* = 10) and healthy control animals (*n* = 10).

### 2.2. Sample Collection

Seven to 0 days prepartum, swabs of vaginal secretions were obtained using a double-guarded sampling system (uterus culture swab with introduction pipette, Minitube, Tiefenbach, Germany) as previously described [20]. Sampling was repeated in animals that did not calve 7 days after initial sampling. Briefly, a fecal swab was obtained by swabbing feces collected per rectum (sterile swab BD CultureSwab, Becton Dickinson Allschwil, Switzerland). Then, endometrial swabs were obtained 1, 4, 7, 10, and 21 DPP using double-guarded swabs to prevent contamination. Briefly, the guarded swab was introduced into the vagina and then guided through the cervix by transrectal manipulation [20]. Swabs were obtained from the uterus (uterine body). Within the uterine body lumen, the inner brush was pushed through the outer guard and rotated 3–5 times against the mucosa. The swab was then retracted into the inner cover and withdrawn. The swabs were then capped and placed on ice before transportation to the lab. At the lab, the swabs were stored frozen at −20 °C.

### 2.3. Sequencing of Microbiomes

Genomic DNA was extracted from the swabs using QIAamp DNA (Qiagen, Hilden, Germany) (vaginal, endometrial) and QIAamp DNA Stool (fecal) Mini Kits, respectively, according to the manufacturer’s instructions. Quantus Fluorometer and the QuantiFlour^®^ Double DNA (dsDNA) system (Promega, Madison, WI, USA) kit was used to determine DNA yields from the samples. Bacterial 16S rRNA V4 regions in the DNA samples were amplified using the 515F and 806R primer combination [21] and subjected to Illumina sequencing at StarSEQ (Mainz, Germany, https://www.starseq.com/ (accessed 30 March 2023)). Sequence data (deposited under Bioproject number PRJNA914879) was processed and analysed in R (R-project-org) using the phyloseq package version 1.40.0 [22]. Read quality filtering, error correction, dereplication, merging, and construction of an operational taxonomical unit (OTU) table was performed in the Dada2 package version 1.24.0 using the settings recommended by the package authors [23]. The 16S rRNA amplicon sequences were annotated using the Silva SSU database version 138.1 and assigned to OTUs with a 97% similarity cut-off. Diversities were calculated using the vegan package 2.6-2 [24], and phylogenetic relations were calculated using the ape package [25]. The alpha diversity of the samples was estimated using the observed OTUs and Shannon diversity metrics, and the statistical significance of the differences was assessed using Kruskall-Wallis tests. Beta diversity was estimated using the Bray-Curtis dissimilarity index. Permutational multivariate analysis of variance (PERMANOVA) tests were used to test for differences in microbial beta diversity between the helathy (*n* = 10) and metritis (*n* = 10) sample groups. Differential OTU abundances between healthy and metritis animals were calculated using the EdgeR normalization [26]. A cluster dendrogram based on the averaged taxa presence and absence composition for cows in each group was generated using CLC genomics (Qiagen, Prismet, Aarhus, Denmark).

### 2.4. Statistical Analysis

Clinical parameters were evaluated using the Mann-Whitney test. Differences in metritis scores were then analyzed using ANOVA for repeated measures and calculated using the Fisher’s post hoc test with a statistical program (StatEL, Paris, France) in Excel (Microsoft, Wallisellen, Switzerland). Significance was set at *p*-value < 0.05. One-way Analysis of similarities (ANOSIM) was performed between multiple groups based on the Bray-Curtis distance metric with 9999 permutations.

Univariate Repeated-measures ANOVA was performed on both H and M groups between all time points. Analyses were performed using R version 4.2.2, utilising ‘vegan’ and ‘heplots’ packages. LDA Effect Size (Lefse) [27] was performed on microbial abundance data for healthy and metritis cattle. Lefse uses a non-parametric factorial Kruskal-Wallis sum-rank test to detect features with significant differential abundance with respect to the class of interest, followed by pair-wise tests of sub-classes using the Wilcoxon rank-sum test.

## 3. Results

### 3.1. Clinical Findings

Healthy animals were on average 5.4 years old and calved after an average gestation length of 283 days. Compared to healthy controls, metritis animals had an average age of 4.6 years and a gestation length of 282 days. Metritis animals showed a significantly higher average vaginal discharge score and rectal temperature compared to healthy animals (*p* < 0.0001 and *p* < 0.05, respectively). The course of vaginal discharge scoring over the study period also differed significantly between the two groups (*p* < 0.0001; Figure 1). Compared to healthy animals, the mean vaginal discharge scores of metritis animals were statistically significantly higher (*p* < 0.01) on 7, 10, and 21 DPP.

### 3.2. Microbiome Comparison

#### 3.2.1. Sequencing Results and Overall Composition of Vaginal, Endometrial, and Fecal Microbiomes

Utilizing bacterial 16S amplicon sequencing the microbiome compositions of prepartum vaginal and fecal as well as postpartum endometrial samples of metritis (*n* = 10) and healthy (*n* = 10) animals were determined. Of the 140 samples (70 metritis and 70 healthy) sequenced, 7674 to 285,339 sequence reads per sample were obtained. Post filtering the sequences were assigned to 5200 OTUs that belonged to 45 different phyla and 625 genera. Overall, Proteobacteria, Bacteroidetes, Firmicutes, Fusobacteria, and Actinobacteria were the predominant phyla detected among the sequenced fecal, vaginal, and endometrial microbiomes (Appendix A). In fecal and vaginal microbiomes Bacteroidetes, Firmicutes, and Proteobacteria predominated, whereas the order of the most predominant phyla in uterine microbiomes varied depending on the uterine disease condition and the examined time point postpartum (see Appendix A). Analysis of similarity (ANOSIM) for comparison of microbial communities in healthy and metritis animals detected significant (*p* = 0.001; see Appendix A) differences in microbiome taxonomic composition between the different postpartum time points. This was further interrogated using univariate repeated-measures ANOVA assuming sphericity, which showed there were significant differences in taxonomic composition between healthy (*p* = 0.0013) (see Appendix A) and metritis (*p* = 0.00016) animals (see Appendix A). To characterise the differences between these groups, Lefse was performed, however, possibly due to natural biological variability between individual animals in each group and the small sample sizes, no statistically significant differences between the groups were discovered.

#### 3.2.2. Comparison of Vaginal and Fecal Microbiomes Prepartum

Examining samples collected from cows prior to delivery revealed no differences in microbial species richness and alpha diversity between the fecal and vaginal microbiomes of metritis and healthy animals (Figure 2A,B). Using taxonomic composition-based clustering, the fecal microbiomes of the examined animals were less compositionally distinct than vaginal microbiomes within and between the two groups (Figure 2C,D). Meanwhile comparing the overall core taxa distributions between the two groups showed that there were more variably distributed genera between the vaginal than fecal microbiomes of the two animal groups (Appendix A). A comparison of the relative abundance of the topmost abundant phyla (*n* = 5) and genera (*n* = 10) showed that fecal and vaginal microbiomes are similar taxonomically between metritis and healthy animals. Despite this, the sequences from the phyla Fusobacteria, and genera such as *Bacteroides*, *Staphylococcus*, *Histophilus* and *Escherichia-Shigella*, were significantly more abundant in vaginal and fecal microbiomes of metritis animals (Figure 2E,F and Appendix A).

#### 3.2.3. Comparison of Postpartum Endometrial Microbiomes

Endometrial microbiomes detected at the beginning (1 DPP, 4 DPP) and end (21 DPP) of the sampling period presented similarly in microbial species richness and alpha diversity levels between metritis and healthy animals (Figure 3A,B). Compared to healthy animals, endometrial microbiomes of metritis cows at 7 and 10 DPP were significantly lower in microbial species richness and alpha diversity (Figure 3A,B). Grouped according to similarities in an averaged group (Figure 3C and Appendix A) as well as individual (Figure 4A) cow taxonomic compositions, the 1, 4, and 21 DPP endometrial microbiomes formed no distinctive clusters, whereas the 7 and 10 DPP uterine microbiomes formed distinct clusters according to the uterine disease status. Based on averaged taxonomic composition, both 7 and 10 DPP uterine microbiomes clustered closely within each group but formed separated clusters between the metritis and healthy microbiomes (Figure 3C and Appendix A). The 7 and 10 DPP uterine microbiomes showed the highest compositional similarity between individual cows in each group and represented the time points at which the highest compositional shift between metritis and healthy endometrial microbiomes was observed (Figure 4A). Taxonomic composition differences were further analyzed by comparing the total number and distribution of all the bacteria genera present in metritis and healthy endometrial microbiomes at different times postpartum (shown on Venn diagrams, Figure 5). Metritis endometrial microbiomes differed from healthy in both the total number and distribution of the overall bacteria genera found at different times postpartum. The greatest discrepancies in the total number and distribution of genus taxa between the animals were in endometrial microbiomes at 7 and 10 DPP (Appendix A). In addition to revealing that there was an overall lower number of genus taxa detected in metritis animals, this comparison also revealed that animals from the two groups had the lowest numbers of commonly shared genera (Appendix A). Comparing the five topmost abundant phyla on the other hand showed that endometrial microbiomes of metritis animals declined in Proteobacteria (51 to 6%) while gradually increasing in both Actinobacteria (2 to 24%) and Bacteroidetes (6 to 8%) relative abundances as postpartum time progressed from 1 to 10 DPP (Figure 4B). In healthy animals, on the other hand the Proteobacteria abundance declined modestly (63 to 39%), whereas Bacteroidetes (8 to 4%) abundance decreased, and there was a significantly lower increase of Actinobacteria (4 to 7%) abundance during this period (Figure 4B). The relative abundances of Firmicutes and Fusobacteria similarly fluctuated between 1 and 10 DPP for animals in both groups. Both phyla were however more abundant in metritis than healthy endometrial microbiomes except on 10 DPP, where there were similar Fusobacteria abundances detected for both groups. Taxa relative abundances examination at the genus level showed that the Proteobacteria abundance declined between 1 and 10 DPP within healthy animals largely due to decreasing abundance of an unclassified Proteobacteria genus (OTU1941) (Figure 4C). Increasing Actinobacteria and Bacteroidetes abundances were associated with rising *Trueperella*, *Bacteroides*, and *Porphyromonas* abundances in metritis animals (Figure 4C). Reduced Proteobacteria relative abundances between 1 and 4 DPP in metritis cows were replaced by increased Fusobacteria and Actinobacteria (4 DPP only) levels, whose relative abundances were significantly higher than those in healthy animals. The 7 and 10 DPP uterine microbiomes showed the most similar taxonomic profiles between the different postpartum days in both groups. Relative to healthy animals, the 7 and 10 DPP metritis animals were less abundant in Proteobacteria but higher in Firmicutes, Bacteroidetes, and Actinobacteria abundances at these timepoints (Figure 4B). *Trueperella*, *Bacteroides*, *Enterococcus*, *Porphyromonas*, *Escherichia-Shigella*, and *Histophilus* were among the genera that displayed higher mean relative abundances within metritis endometrial microbiomes at 7 and 10 DPP (Figure 4C and Appendix A). The unclassified Proteobacteria genus OTU1941, in contrast, dominated in healthy endometrial microbiomes and was significantly more abundant than in metritis animals (Figure 4C and Appendix A). Compared to healthy cows, the 1 and 4 DPP endometrial microbiomes of metritis animals also displayed a higher abundance of other Proteobacteria genera including *Histophilus* (Figure 4C). Endometrial microbiomes in metritis animals on the other hand had rebounded and stabilized on 21 DPP as their microbial species diversity (Figure 3A,B) and relative abundances (Figure 4B,C) of predominating phyla were like those in healthy cows. Despite this, 21 DPP microbiomes between cows of the two groups still maintained some differences in abundance and distribution of various bacterial phyla and genera (Figure 4B,C, Appendix A). In addition to variable distribution for several of the less abundant genera, the 21 DPP metritis endometrial microbiomes, compared to healthy animals, were more abundant in several shared genera including *Trueperella*, *Bacteroides*, *Porphyromonas*, *Prevotella*, and *Histophilus*.

#### 3.2.4. Distribution of Bacterial Taxa between Vaginal, Endometrial, and Fecal Microbiomes

The impacts of metritis on endometrial microbiome composition were also probed by comparing the distribution of bacterial genera between metritis and healthy core endometrial microbiomes postpartum. These were comprised of bacterial genera detected among cows of each group at different time points during the postpartum. period Based on these criteria, core postpartum endometrial microbiomes of 45 and 78 genera, respectively, were identified in metritis and healthy endometrial microbiomes (Figure 5A,B and Appendix A). Nine and 42 genera were found unique to the core endometrial microbiomes of postpartum metritis and healthy animals, respectively (Figure 5C and Appendix A). Genera such as *Fusobacterium*, *Mycoplasma*, and *Enterococcus* were unique to the metritis core, whereas *Prevotella* and *Lactobacillus* were among the genera unique to the healthy core endometrial community. As much as 76% (34/45) of the metritis and 68% (53/78) of the healthy postpartum core endometrial microbial communities were common with their corresponding core prepartum fecal and vaginal microbiomes (Figure 6A,B). Eight of the genera including *Enterococcus* and *Escherichia-Shigella* were only shared between core vaginal, endometrial and fecal microbiomes in metritis animals. There were 10 and 22 genera, respectively, within metritis and healthy core endometrial microbiomes postpartum that were only shared with the corresponding prepartum core vaginal microbiomes (Figure 6A,B). These genera included 5 and 17 genera, respectively, that were exclusive to metritis and healthy animals (Figure 6D; Appendix A).

## 4. Discussion

The natural genital microbiomes are beneficial to the host through different mechanisms including production of protective biofilms as well as other host defense molecules. Both endometrial and vaginal microbiomes are critical in the development of endometritis, which is an important cause of infertility in dairy cattle. Bacterial 16S rRNA sequencing previously showed that the normal vaginal microbiome comprises bacterial phyla Bacteroidetes, Fusobacteria, and Proteobacteria [2]. In addition, it is known that postpartum endometrial microbiomes in cows have important roles in the modulation of uterine health and the development of metritis [2]. Thus, metritis in cows is commonly associated with a dysbiosis that causes less complexity and altered taxonomic composition of endometrial microbial communities [2,10,17,28,29]. Compared to healthy animals, metritis cows harbour notable endometrial taxonomic compositional changes of their microbial communities showing more Fusobacteria, Bacteroidetes, and Actinobacteria, but less Proteobacteria abundances [2,10,11,15,22,23,24,25,26,27,28,29,30,31,32,33,34,35,36,37,38]. *Fusobacterium*, *Trueperella*, *Bacteroides*, *Escherichia-Shigella*, *Histophilus*, and *Porphyromonas* are among bacterial genera that are reported to increase their abundance in metritis compared to healthy cows [2,32,34]. Notably, these genera include species of specific endometritis pathogens such as *Escherichia coli*, *Trueperella pyogenes*, *Fusobacterium necrophorum*, *Prevotella melaninogenica*, and *Porphyromonas levii¸* which have been isolated using culture-dependent approaches from metritis infections [7,29,39,40].

Bacteria deriving from fecal and vaginal microbiomes contribute to the post-calving uterine microbial communities in cows [2,9,41,42]. Bacteroidetes, Fusobacteria, Firmicutes, Proteobacteria, Actinobacteria, and Tenericutes sequences were previously found to predominate among the fecal and vaginal microbiomes detected in cows [9,41,42,43,44,45]. Similar composition of predominating bacterial phyla was determined in the fecal and vaginal microbiomes found prior to calving in metritis and healthy cows examined in this study. Previously, pre-calving vaginal microbiomes in cows diagnosed with metritis were reported to contain higher Proteobacteria loads than those of healthy cows [33]. In the current study, compared to healthy animals, metritis cows also displayed some taxonomic compositional differences. In particular, we found that fecal and vaginal microbial communities in cows that developed metritis were higher in an abundance of some of the bacterial taxa that are known to include metritis infection-associated pathogenic bacterial species such as the phyla Fusobacteria and the genera *Escherichia-Shigella* [2,6,13]. Metritis cows further differed from healthy cows in the occurrence and distribution of various other less abundant bacteria taxa. Taken together, some of these microbial community differences may have contributed to postpartum endometrial microbiomes in ways that impacted uterine health or the development of metritis postpartum.

Overall, our analysis showed that the endometrial microbial communities are similarly dynamic within both animal groups. In line with previous observations, we also found that endometrial microbial communities in metritis cows shifted their structure and composition when compared to healthy animals. Lower Proteobacteria and higher Fusobacteria, Bacteroidetes, and Actinobacteria abundances in uterine microbial communities have been associated with metritis development whereas increased Proteobacteria and Tenericutes abundances have been associated with a healthy uterus [2]. In our study, the metritis cows similarly showed an overall gradual depletion of Proteobacteria, which was replaced through increasing Fusobacteria, Firmicutes, Bacteroidetes, and Actinobacteria populations between 1 and 10 DPP. Although Fusobacteria and Firmicutes’ relative abundances similarly fluctuated within both groups both phyla in most cases were more abundant within metritis animals. Differences of varying magnitude in endometrial microbial community composition between metritis and healthy cows were associated with changes in distribution and relative abundance of different bacteria taxa detected. Statistically significant changes with respect to species richness and alpha diversity of uterine microbial communities were however only apparent at 7 and 10 DPP between cows from the two groups. Endometrial microbial populations on 7 and 10 DPP were most distinct between healthy and metritis animals. During this period, the increasing divergence in the endometritis score observed between the metritis and healthy animals is in line with many incidences of metritis detected at 5 and 7 DPP as previously reported [46].

## 5. Conclusions

A deeper understanding of endometrial microbiota structure and compositional dynamics will pave the road for a new era in research for diagnosing uterine disease, improving fertility, and enhancing the productivity of dairy animals. Our results suggest that microbial communities found in feces and the vagina prepartum may play a key role in composing the endometrial microbiome postpartum. Thus, an improved understanding of the relationship between pre-calving fecal and vaginal microbiomes and postpartum endometrial microbial communities would open possibilities of predicting uterine disease in cows based on pre-calving fecal and vaginal microbiomes. Harnessing this knowledge may revolutionize monitoring and treating uterine disease in dairy animals. More research is needed in order to develop robust and comprehensive protocols incorporating pre- and probiotic products that support the proliferation of ‘good’ endometrial microbiomes.

## Figures and Tables

**Figure 1 animals-13-01221-f001:**
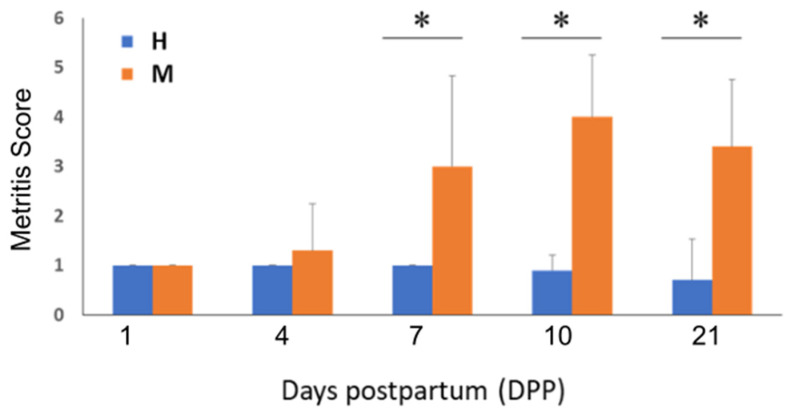
Comparison of endometritis score changes during the study period. Bar graph depicting averaged metritis scores for the metritis (M) and healthy (H) animal groups from 1 to 21 DPP. * Indicates statistically significant (*p* < 0.01) differences between metritis and healthy animals’ endometritis scores.

**Figure 2 animals-13-01221-f002:**
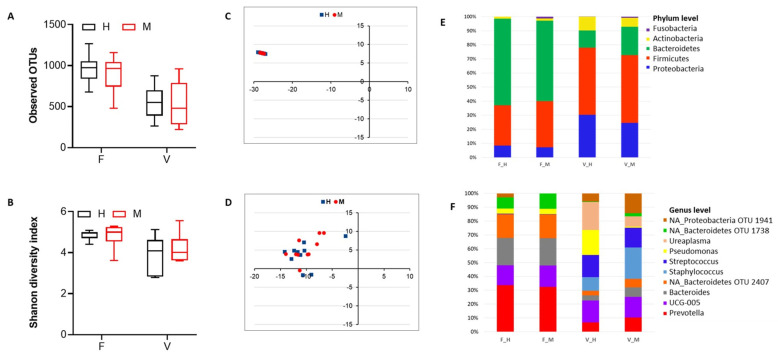
Comparison of microbial diversity and taxonomic composition between fecal (F) and vaginal (V) microbiomes found in metritis (M) and healthy (H) animals prepartum. Box plots showing microbial community species richness and alpha diversity based on observed OTUs (**A**) and Shannon diversity index (**B**), respectively. NMDS plots based on Bray-Curtis dissimilarity distances between pre-calving fecal (**C**) and vaginal (**D**) microbiomes of M and H cows. Individual points represent single cows (*n* = 10 for each group). Blue—H cows; Red—M cows. Bar graphs showing averaged relative abundances and distribution for the top 5 phyla (**E**) and top 10 genera (**F**). Unclassified genera are denoted by NA. Relative abundances were rescaled to account for only the 5 (phyla) and 10 (genera) most abundant phyla. Taxonomic profiles for the top 50 taxa are provided in Appendix A. F—fecal; V—vaginal; F_H—fecal healthy, F_M—fecal metritis, V_H—vaginal healthy, V_M—vaginal metritis.

**Figure 3 animals-13-01221-f003:**
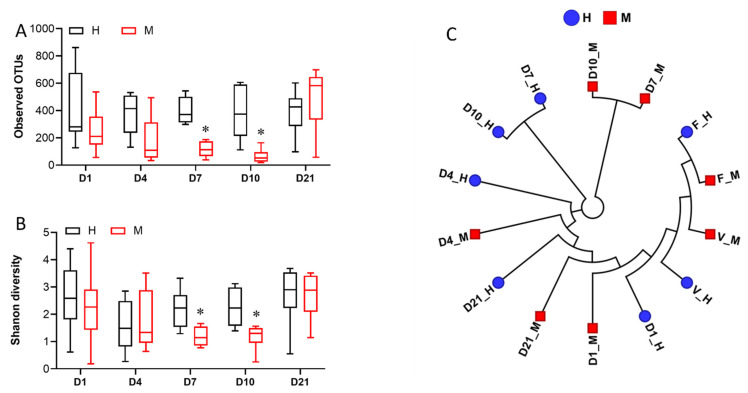
Comparison of microbial species diversity and taxonomic composition between postpartum uterine microbiomes in metritis (M) and healthy (H) animals. Box plots showing microbial community species richness and alpha diversity based on (**A**) observed OTUs and (**B**) Shannon diversity index, respectively. * Indicates statistically significant (*p* < 0.05; Kurskall Wallis test) differences between M and H animals’ alpha diversities. (**C**) Hierarchal clustering cladogram showing similarities between M and H uterine microbiomes based on averaged taxa presence and absence composition of uterine microbiomes of animals from each group.

**Figure 4 animals-13-01221-f004:**
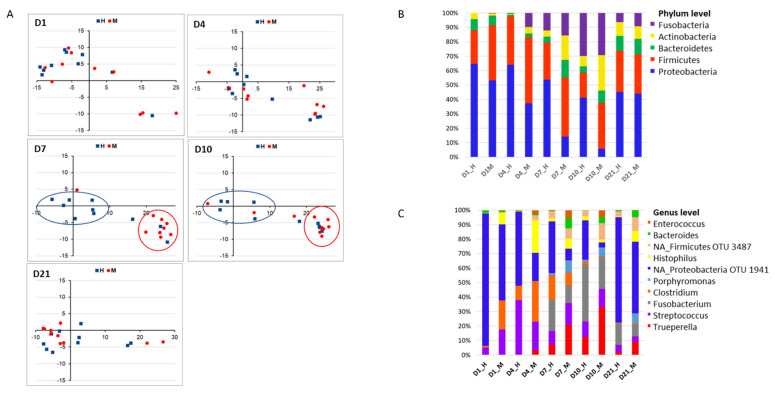
Comparison of the taxonomic composition of postpartum uterine microbiomes between healthy (H) and metritis (M) animals. (**A**) NMDS plots based on Bray-Curtis dissimilarity distances between postpartum uterine microbiomes of individual cows for each time point. Individual points represent single cows (*T* = 10 for each group). Blue—healthy cows; Red—metritis cows. Bar graphs showing averaged relative abundances and distribution for the top 5 phyla (**B**) and top 10 genera (**C**). Unclassified genera are denoted by NA. Relative abundances were rescaled to account for only the 5 (phyla) and 10 (genera) most abundant phyla.

**Figure 5 animals-13-01221-f005:**
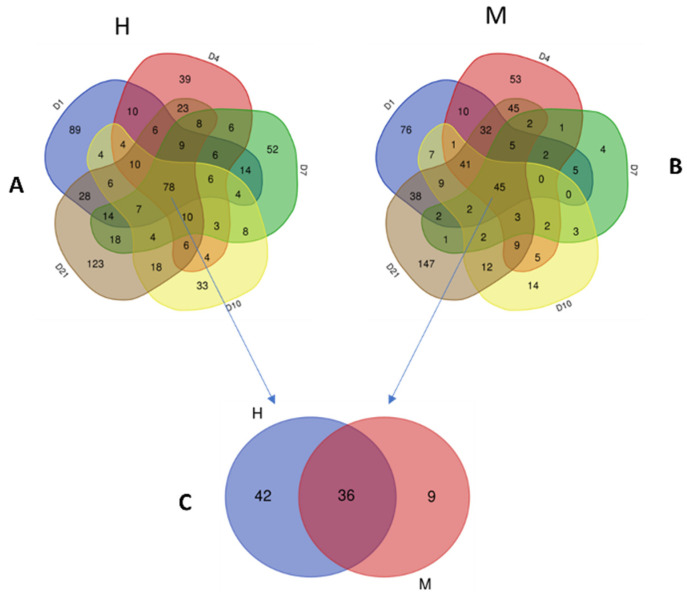
Venn diagrams comparing genus OTU composition of uterine post-partum D1-D21 microbial communities between healthy (H) and metritis (M) animals. OTUs found at all postpartum time points in at least one cow per group were determined (**A**,**B**) and compared (**C**) between H and M cows. 78 and 45 OTUs, respectively, were found in postpartum microbiomes of (**A**) H and (**B**) M cows at all time points. Comparing these OTUs showed that 36 were shared, while 42 and 9 OTUs were exclusive to H and M cows, respectively. Details of these genera are provided in Appendix A.

**Figure 6 animals-13-01221-f006:**
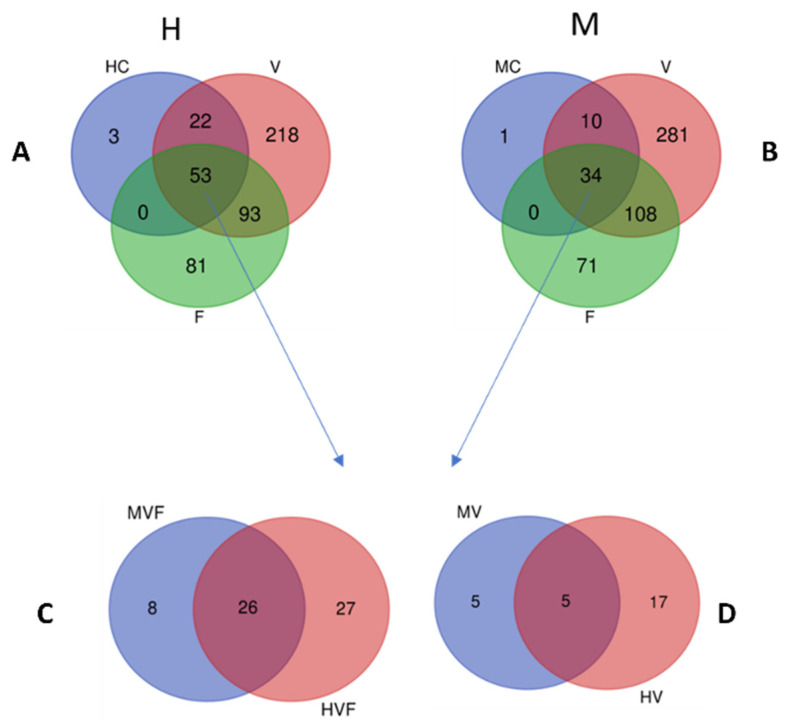
Venn diagrams show OTU distributions between core postpartum uterine microbiomes (OTUs detected in at least one cow per group at all postpartum time points) and the core prepartum vaginal (V) and fecal (F) microbiomes in (**A**) healthy (HC) and (**B**) metritis (MC) cows. Comparison of core healthy (H) and metritis (M) postpartum uterine microbiome bacteria taxa genera that are shared (intersections of HC, V, and F in (**A**) and MC, V and F in (**B**)) with (**C**) both core prepartum vaginal and fecal microbiomes, and with (**D**) the core vaginal microbiome only.

## Data Availability

The data sets generated for this study are accessible in NCBI under bioproject number PRJNA914879.

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
