# Peer review of "Interrogating the Diversity of Vaginal, Endometrial, and Fecal Microbiomes in Healthy and Metritis Dairy Cattle"

_animals, 2023, doi:10.3390/ani13071221_

Round 1

Reviewer 1 Report

The paper under review aimed to interrogate vaginal, endometrial, and faecal microbiomes in dairy cows and determine whether these profiles predict risk of uterine disease postpartum. Thes study showed that compared to health animals the pre-calving fecal and vaginal microbiomes of metritis animals were more abundant in sequences from the phylum Fusobacteria and the bacterial genera such as Escherichia-Shigella and Histophilus. Metritis cows showed an overall gradual depletion of  Proteobacteria and increasing Fusobacteria, Firmicutes, Actinobacteria and Bacteroidetes abundances in the endometrium between 1 and 10 days postpartum. There was high taxonomic similarity detected between postpartum endometrial microbiomes and the prepartum vaginal and fecal microbiomes.

In general, the study confirmed that bacteria deriving from faecal and vaginal microbiomes contribute to the post-calving uterine microbial communities in cows, and that there are compositional differences in pre-calving fecal and vaginal microbiomes between cows with metritis and healthy cows. In line with previous observations, the study showed that lower Proteobacteria and higher Fusobacteria abundances in uterine microbial communities have been associated with metritis.

The manuscript is well written, however, there is still room for improvement.

In Materials and Methods, please state whether the cows were from one herd or from different herds, whether calving assistance was provided and whether there were placental retentions.

I am very doubtful about the authors' use of the term “endometritis score” to assess the character of vaginal discharge in cows with metritis. These are different disease entities. According to Sheldon (2008), who is cited by the authors, clinical endometritis is characterised by the presence of purulent (>50% pus) uterine discharge detectable in the vagina 21 days or more after parturition, or mucopurulent (approximately 50% pus, 50% mucus) discharge detectable in the vagina after 26 days. I therefore recommend using the term “vaginal discharge score”.

Puerperal and clinical metritis were combined together. It would be interesting to know if there were differences in microbiomes between these forms of metritis.

The conclusions are general in nature. Please refer to the possibility of predicting the risk of uterine diseases based on the pre-calving fecal and vaginal microbiomes.

Reviewer 2 Report

Dear authors,

With interest I have reviewed critically your submitted manuscript for publication entitled 'Interrogating the diversity of vaginal, endometrial, and fecal microbiomes in healthy and metritis dairy cattle' which provides insight in the differences/variations of (at both phylum and genus level) microbiomes in Healthy and Metritis cows at different days post partum. All over the paper is scientifically well written and constructed and therefore good to understand including the (supplementary) figures and tables depicted, despite a number typing errors and a list of remarks that do ask clarification for improvement. More information is necessary for the cows included in your study although Figure 1 provides solid an statistically sound comparison on the clinical differences (endometritis score) between the H and M cows finally selected for the study sampling and analysis. All over r Results an d clearly depicted and Discussion/Conclusions are sound and good to understand.

List of remarks/suggestions (per section/lines) to (re)consider for improvement:

- Title: well written and covers the study topic and aim.

- Summary: clear to understand

- Abstract: no comment, covers the study design, results and conclusions/prospects

- Introduction: no comments, leads to the gap/aim of study reflecting literature to support this section

- Material and Methods:

Line 81: more information on the cows is necessary to be included on                the calving procedure and production status; at what day clinical examination was performed; the main issue is about the selection and inclusion of the animals: from 39 animals 19 were excluded receiving antibiotics (AB) meaning clinically severe cows were excluded, correct? 10 M cows were included in the study NOT having AB. My Q: what is the difference between the AB M cows and the non AB M cows? Non AB M cows selected were NOT systemically diseased from metritis (LPS)? Correct?

For endometrial swabs it is NOT mentioned exactly the region of the uterus where the samples were taken, add this information. What about the possible influence of (contamination with) intra uterine fluid?

Explain abbreviations in general (also see the Figures/Tables) but also here 'OTU'.

Results

3.1 these results given illustrate indeed H cows were (significantly) different from M cows selected, proven the clinical scoring is discrimitive. 

line 165: type error 'healthy'

line 177: were not was

lines 193-196: this text does not match with Figure 2 A and B: genus level mentions 'Enterococcus', text mentions Histopholus and Escherichia-Shigella

Lines 203-207 (Figure 4A) seems to provide copy-paste information when compared to lines 210-213: check this again and if yes, rephrase.

Line 206: skip 'Figures 3C and 4A', is double now

Lines 247, 250, 257: add supplementary

Figures and legends:

Figure 1: Y axis Endometritis -> consider Metritis because that is where the study is about

Figure 2: explain all (also V and F) used abbreviations in legend

Figure 3: although used in text, A, B and C are not mentioned, check also in legend for correctness; check again if C. is clearly explained in the legend.

Figure 4: check legend for correctness om A, B, and, not correct now

Figure 5: add supplementary Table S2

Figure 6: improve the legend of this Table, explain more clearly content and abbreviations

References:

Ssound and representative for this research topic presented.
